# Common hemoglobin variants affecting the diagnosis of β-thalassemia: A large cohort data at a single center

Kritsada Singha[1,2], Hataichanok Srivorakun[1], Supawadee Yamsri[1], Attawut Chaibunruang[1], Anupong Pansuwan[1], Yossombat Changtrakul[3], Kanokwan Sanchisuriya[1], Goonnapa Fucharoen[1], Supan Fucharoen[1]*

1 Centre for Research and Development of Medical Diagnostic Laboratories, Faculty of Associated Medical Sciences, Khon Kaen University, Khon Kaen, Thailand, 2 Biomedical Science Research Unit, Faculty of Medicine, Mahasarakham University, Mahasarakham, Thailand, 3 Clinical Microscopy Unit, Srinagarind Hospital, Khon Kaen University, Khon Kaen, Thailand

* supan@kku.ac.th

## Abstract

### Background

Many globin chain variants were interpreted as β-thalassemia or hereditary persistence of fetal hemoglobin (HPFH) at routine investigation. We described this in a large cohort of Thai subjects.

### Methods

Hematological data of 43,414 subjects encountered at our thalassemia diagnostic center from January 2013 to July 2025 were reviewed. A total of 372 subjects with hemoglobin (Hb) variants were selectively recruited with leftover DNA specimens for further analysis. Hb analysis was done using high-performance liquid chromatography (HPLC) or capillary electrophoresis. β-globin gene mutations were identified using PCR and related techniques.

### Results

Among 372 subjects recruited, a total of 21 different Hb variants were found. The levels of Hb $A_2$, Hb F, Hb variants, and DNA diagnostic requests were recorded. Hb variants with normal Hb $A_2$ and Hb F levels were requested for β-thalassemia or HPFH in 12.3% (95% CI = 7.9–16.8). This error was increased in Hb variants with Hb $A_2$ ≥ 3.6% [50.6% (95% CI = 39.8–61.4) and odds ratio of 7.3 (3.6–13.8)] and Hb F ≥ 5.0% [52.2% (95% CI = 37.7–66.6) and odds ratio of 7.8 (3.6–16.7)], especially in those with Hb $A_2$ ≥ 3.6% and Hb F ≥ 5.0% [71.9% (95% CI = 56.3–87.5) and odds ratio of 18.2 (7.1–48.9)]. Hbs Hope, Tak, Cook, C, Lepore, and Q-Thailand were found to be associated with high proportions of the error.

**Data availability statement:** All relevant data are within the manuscript files.

**Funding:** This study was financially supported by the Fundamental Fund of Khon Kaen University, under the National Science, Research and Innovation Fund (NSRF), Thailand, to SF (Contract ID: FF2569 KKU), and Genomics Thailand, the Health System Research Institute (Contract ID: HSRI 68-049) to Dr. Kritsada Singha. We thank Mahasarakham University Development Funds, Mahasarakham University, Thailand, for supporting the fund for an international research presentation. This funding was received by Dr. Kritsada Singha. The funders had no role in study design, data collection and analysis, decision to publish, or preparation of the manuscript.

**Competing interests:** The authors have declared that no completing interests exist.

## Conclusions

Hb variants with increased Hb $A_2$ levels and/or co-migration with Hb F are associated with a high proportion of mis-interpretation of β-thalassemia in routine practice. Combining Hb analysis using two different methods, and molecular analysis should help improve laboratory interpretation of the cases.

## Introduction

The inherited hemoglobin (Hb) disorders are the commonest human monogenic disease, broadly divided into structural Hb variants, thalassemia, and hereditary persistence of fetal hemoglobin (HPFH) [1,2]. Structural Hb variants are a group of genetic disorders characterized by structural changes in the globin chains, whereas thalassemia is characterized by the absence or reduced synthesis of particular globin chains. High Hb F determinants are genetic defects associated with increased expression of Hb F in adult life. In areas with a high prevalence of thalassemia and hemoglobinopathies, genetic interactions of these defects could lead to diverse heterogeneity and several complex thalassemia syndromes. Accurate diagnosis is essential for providing genetic counseling, appropriate management, and facilitating a prevention and control program in the region [1–3].

Structural Hb variants have been continually described worldwide, with more than 1,400 types. The majority of known Hb variants have arisen by substitutions, involving the β-globin and α-globin chains, the major adult Hb components (Hb A) [4]. Although most Hb variants are benign, some would cause clinical symptoms, such as unstable Hbs, hemolytic anemia, thalassemia phenotype, altered oxygen affinity, methemoglobin, and abnormal polymerization. Other associated pitfalls include interference with HbA1c measurement and unnecessary prenatal diagnosis [1–8]. In Thailand, high-performance liquid chromatography (HPLC) and capillary electrophoresis are widely utilized for Hb analysis [3,5–7]. More than 50 Hb variants have been described. Among them, Hb Hope, Hb Tak, and Hb Q-Thailand are the most common ones [5–7]. In routine diagnosis of thalassemia and hemoglobinopathies at our center, many subjects with Hb variants were associated with elevated Hb $A_2$ levels, co-migrated with Hb F, and were frequently interpreted as β-thalassemia or HPFH. Here, we demonstrate these Hb variants affecting the diagnosis of β-thalassemia in the large cohort of Thai subjects, along with recommendations for providing accurate interpretation.

## Materials and methods

### Subjects and hematological analyses

The study protocol of this research was approved by the Institutional Review Board (IRB) of Khon Kaen University, Thailand (HE682074). Informed consent was waived because this is a study of existing data or biological specimens without further prospective data collection from or direct interactions with the subjects. Among 43,414 subjects referred to our center for investigation of thalassemia and

hemoglobinopathies from 01/01/2013–31/07/2025, a total of 372 subjects with Hb variants in heterozygous form, together with their leftover specimens, were selectively recruited. All subjects with Hb variants but Hb E were included. This is because it has been known that Hb E, the most common Hb variant in Southeast Asia, is associated with elevated Hb $A_2$ and can be easily diagnosed on Hb analysis, not affecting the diagnosis of β-thalassemia [2,3]. The retrospective data and leftover DNA were accessed for research after ethical approval on 01/04/2025 and analyzed until 31/08/2025. Hb analysis was performed by using capillary electrophoresis (Capillarys II Flex Piercing, Sebia, France) or high-performance liquid chromatography (HPLC) (Variant™, Bio-Rad Laboratory, Hercules, CA, USA). Hb $A_2$ and Hb F levels, DNA analysis requirements for the first visit by physicians or medical staff, and a definitive diagnosis by molecular analysis were recorded. Normal Hb $A_2$ and Hb F levels were set at Hb $A_2 < 3.6\%$ and Hb F $< 5.0\%$ because Hb $A_2 \geq 3.6\%$ were the cut-off for diagnosis of β-thalassemia carriers, and Hb F $\geq 5.0\%$ were associated with HPFH or β-thalassemia diseases [3,8].

### DNA analysis

Known Hb variants, β-thalassemia, and high Hb F determinants found in Thailand were identified by PCR-based methods as described previously [3,5,9]. Uncharacterized Hb variants were examined by whole β-globin gene sequencing using an ABI PRISM™ 3730 XL analyzer (Applied Biosystems, Foster City, California, USA) and Barcode-tagged DNA sequencing (BTSeq™, Celemics, Korea).

### Statistical analysis

Statistical analysis was performed using Stata software version 18 (Stata Corp, Texas, USA). Hb $A_2$ and Hb F levels were reported as mean ± standard deviation. The proportion of DNA analysis requests was described as a percentage with 95% confidence intervals. The odds ratio of the misinterpretation of Hb variants with high Hb $A_2$ and/or Hb F levels was compared to Hb variants with normal Hb $A_2$ and Hb F levels. The odds ratio of Hb variants interpreted as β-thalassemia or HPFH in Hb variants with high Hb $A_2$ and/or Hb F levels was compared to those with normal Hb $A_2$ and Hb F levels.

### Results

Among 372 subjects recruited, 21 different Hb variants were detected, as summarized by year in Fig 1. Hb variants were classified based on the levels of Hb $A_2$ & Hb F into four groups: normal Hb $A_2$ and Hb F levels (Hb $A_2 < 3.6\%$ and Hb F $< 5.0\%$), Hb $A_2 \geq 3.6\%$, Hb F $\geq 5.0\%$, and (Hb $A_2 \geq 3.6\%$ and Hb F $\geq 5.0\%$). DNA analysis requirements at the first visit by physicians in each group were recorded. As shown in Table 1, most Hb variants with normal Hb $A_2$ and Hb F levels (Hb $A_2 < 3.6\%$ and Hb F $< 5.0\%$) were requested for DNA analysis of Hb variants [87.7% (95% CI = 83.2–92.1)], whereas some of them were requested as β-thalassemia or HPFH [12.3% (95% CI = 7.9–16.8)]. Interestingly, the request for β-thalassemia or HPFH was increased in those with Hb variants with Hb $A_2 \geq 3.6\%$ [50.6% (95% CI = 39.8–61.4)]. The error was also observed in the Hb variants, with Hb F $\geq 5.0\%$ in 52.2% (95% CI = 37.7–66.6) of the cases. The dramatic increase in error [71.9% (95% CI = 56.3–87.5)] was observed in Hb variants with both Hb $A_2 \geq 3.6\%$ and Hb F $\geq 5.0\%$. Hb variants with Hb $A_2 \geq 3.6\%$, Hb F $\geq 5.0\%$, and Hb $A_2 \geq 3.6\%$ and Hb F $\geq 5.0\%$ had an odds ratio of 7.3 (3.6–13.8), 7.8 (3.6–16.7), and 18.2 (7.1–48.9) times for the error higher than Hb variants with normal Hb $A_2$ and Hb F levels, respectively.

Table 2 lists the Hb variants identified with normal and abnormal Hb $A_2$ and Hb F levels, and the number of DNA analysis requests. They were classified into β-, hybrid, and α-globin variants. Hb Hope (20.2%), Hb Tak (18.0%), and Hb Q-Thailand (12.4%) were the three most common Hb variants found in this study. We found that subjects with Hb Hope, Hb Tak, Hb Cook, Hb C, Hb Raleigh, Hb Lepore, and Hb Q-Thailand with high Hb $A_2$ level and/or co-migrated with Hb F had high proportions of requests for β-thalassemia or HPFH. In contrast, although Hb S was associated with high Hb $A_2$ and/or Hb F levels, it was associated with low proportions of the error. With low sample numbers, the effects of Hb J-Wenchang-Wuming (n = 3), Hb J-Kaohsiung (n = 2), Hb Thailand (n = 2), Hb Raleigh (n = 1), Hb Grey Lynn (n = 1), and Hb J-Buda (n = 1) could not be ascertained.

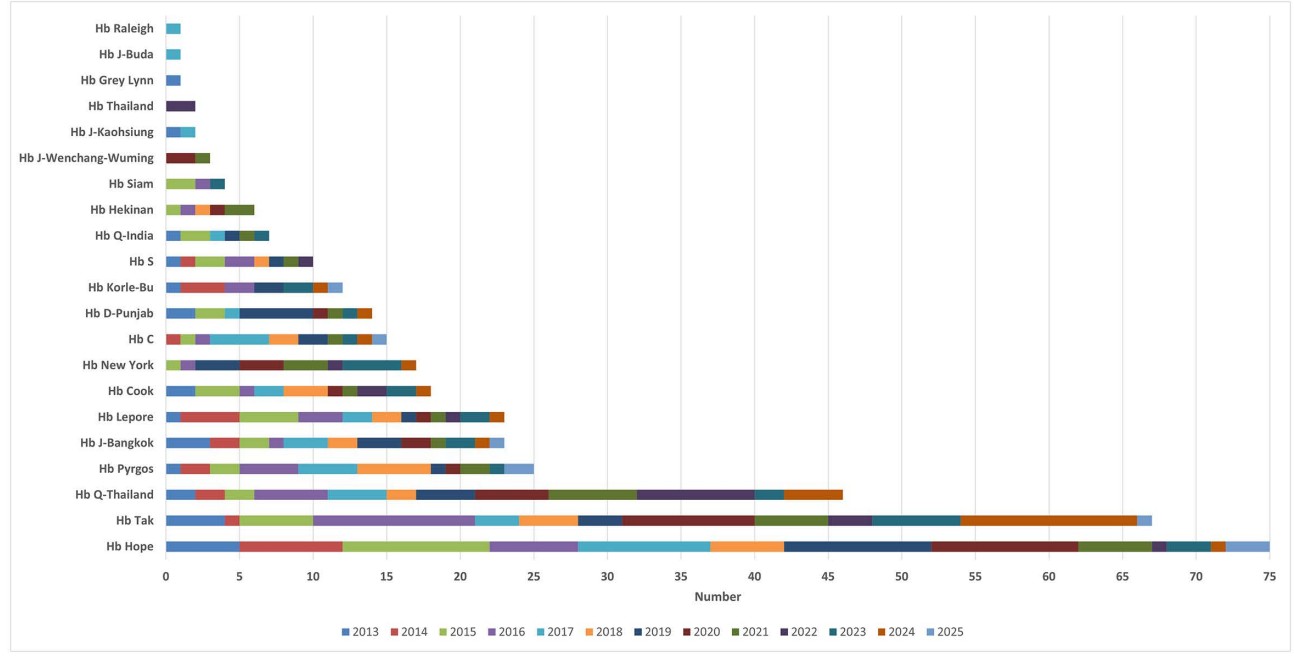

**Fig 1. Numbers of Hb variants encountered each year during 2013-2025.**

**Table 1. Comparison of DNA analysis requests for β-thalassemia or HPFH and Hb variants with normal and abnormal Hb A$_2$ and Hb F levels.**

| Groups of Hb variants | n | Hb A$_2$ (%) | Hb F* (%) | Request for Hb variants | | Request for β-thalassemia or HPFH | | Odd ratio (95%CI) |
|---|---|---|---|---|---|---|---|---|
| | | | | n | % (95% CI) | n | % (95% CI) | |
| Hb A$_2$ < 3.6% and Hb F < 5.0% | 211 | 2.6 ± 0.6 | 0.7 ± 1.1 | 185 | 87.7 (83.2-92.1) | 26 | 12.3 (7.9-16.8) | 1 |
| Hb A$_2$ ≥ 3.6% | 83 | 4.2 ± 0.6 | 0.6 ± 1.1 | 41 | 49.4 (38.6-60.2) | 42 | 50.6 (39.8-61.4) | 7.3 (3.6-13.8) |
| Hb F ≥ 5.0%* | 46 | 2.3 ± 0.5 | 31.3 ± 13.7 | 22 | 47.8 (33.4-62.3) | 24 | 52.2 (37.7-66.6) | 7.8 (3.6-16.7) |
| Hb A$_2$ ≥ 3.6% and Hb F ≥ 5.0%* | 32 | 4.5 ± 1.6 | 25.3 ± 9.9 | 9 | 28.1 (12.5-43.7) | 23 | 71.9 (56.3-87.5) | 18.2 (7.1-48.9) |
| Total | 372 | 3.1 ± 1.1 | 6.5 ± 12.8 | 257 | 69.1 (64.4-73.8) | 115 | 30.9 (26.2-35.6) | – |

Abbreviations: CI, confidence interval

*Hb F level and/or Hb variants co-migration with Hb F.

**Fig 2** shows representative Hb analysis using capillary electrophoresis and HPLC of Hb variants with high Hb A$_2$ levels and/or co-migration with Hb F, and a high proportion of requests for β-thalassemia or HPFH, including Hb Hope, Hb Tak, Hb Cook, Hb C, Hb Lepore, and Hb Q-Thailand. Those Hb variants with low proportions of the error, including Hb New York, Hb D-Punjab, Hb Korle-Bu, Hb S, Hb Hekinan, and Hb J-Wenchang-Wuming, are demonstrated in **Fig 3**. Further DNA analysis of the β-globin gene in these subjects identified a β-thalassemia mutation only in one case. This subject was found to be a double heterozygote for Hb Q-Thailand and β-thalassemia who had a total Hb A$_2$ of 6.5% [(Hb A$_2$ (α$_2$δ$_2$) 5.2% and Hb A$_2'$ (α$^{QT}_2$δ$_2$) 1.3%)], as shown in **Fig 2M**.

## Discussion

Next-generation sequencing (NGS) is now playing an important role in mass screening and diagnosis of thalassemia and hemoglobinopathies in some countries, particularly China [10]. However, in Thailand, diagnosis of thalassemia is generally

**Table 2. The request of DNA analysis in each Hb variant identified with normal and abnormal Hb A₂ and Hb F levels.**

| Hb variants | HGVS name | n | Hb A$_2$ (%) | Hb F$^a$ (%) | Abnormal Hb (%)$^b$ | Normal Hb A$_2$ and Hb F [n (%)] | High Hb A$_2$ and/or Hb F* [n (%)] | Request for Hb variants [n (%)] | Request for β-thal or HPFH [n (%)] |
|---|---|---|---|---|---|---|---|---|---|
| **β-globin variants** | | **279 (75.0%)** | **3.2±0.8** | **6.2±13.2** | **37.8±10.0** | **149 (53.4%)** | **130 (46.6%)** | **186 (66.7%)** | **93 (33.3%)** |
| Hb Hope | HBB:c.410G>A | 75 (20.2%) | 3.3±0.8 | 8.4±17.2 | 38.1±5.5 | 35 (46.7%) | 40 (53.3%) | 50 (66.7%) | 25 (33.3%) |
| Hb Tak | HBB:c.440_441dupAC | 67 (18.0%) | 3.9±0.6 | 13.6±15.0 | 28.5±5.8 | 6 (9.0%) | 61 (91.0%) | 26 (38.8%) | 41 (61.2%) |
| Hb Pyrgos | HBB:c.251G>A | 25 (6.7%) | 2.3±0.5 | 3.3±9.6 | 55.3±3.2 | 21 (84.0%) | 4 (16.0%) | 22 (88.0%) | 3 (12.0%) |
| Hb J-Bangkok | HBB:c.170G>A | 23 (6.2%) | 2.3±0.5 | 1.1±1.1 | 47.8±6.2 | 23 (100%) | 0 (0%) | 20 (87.0%) | 3 (13.0%) |
| Hb Cook | HBB:c.398A>C | 18 (4.8%) | 3.7±0.7 | 0.5±0.8 | 33.5±8.0 | 6 (33.3%) | 12 (66.7%) | 11 (61.1%) | 7 (38.9%) |
| Hb New York | HBB:c.341T>A | 17 (4.6%) | 3.0±0.3 | 0.4±0.9 | 43.2±3.3 | 16 (94.1%) | 1 (5.9%) | 15 (88.2%) | 2 (11.8%) |
| Hb C | HBB:c.19G>A | 15 (4.0%) | 3.2±0.7 | 0.2±0.3 | 32.3±3.3 | 11 (73.3%) | 4 (26.7%) | 9 (60.0%) | 6 (40.0%) |
| Hb D-Punjab | HBB:c.364G>C | 14 (3.8%) | 2.8±0.6 | 0.6±0.7 | 36.7±3.8 | 14 (100%) | 0 (0%) | 13 (92.9%) | 1 (7.1%) |
| Hb Korle-Bu | HBB:c.220G>A | 12 (3.2%) | 2.7±0.3 | 0.4±0.6 | 43.7±0.8 | 12 (100%) | 0 (0%) | 9 (75.0%) | 3 (25.0%) |
| Hb S | HBB:c.20A>T | 10 (2.7%) | 3.3±1.0 | 2.4±2.7 | 34.4±10.7 | 4 (40.0%) | 6 (60.0%) | 9 (90.0%) | 1 (10.0%) |
| Hb J-Kaohsiung | HBB:c.179A>C | 2 (0.5%) | 2.3, 4.9 | 0 | 36.9 | 1 (50%) | 1 (50%) | 2 (100%) | 0 (0%) |
| Hb Raleigh | HBB:c.5T>C | 1 (0.3%) | 2 | 47.1 | 47.1$^c$ | 0 (0%) | 1 (100%) | 0 (0%) | 1 (100%) |
| **Hybrid variant** | | | | | | | | | |
| Hb Lepore | NG_000007.3:g.63290_70702del NG_000007.3:g.63632_71046del | 23 (6.2%) | 3.3±2.2 | 3.7±1.9 | 10.0±1.7 | 17 (73.9%) | 6 (26.1%) | 16 (69.6%) | 7 (30.4%) |
| **α-globin variants** | | **70 (18.8%)** | **2.4±1.1** | **9.0±13.3** | **27.0±7.4** | **45 (64.3%)** | **25 (35.7%)** | **55 (78.6%)** | **15 (21.4%)** |
| Hb Q-Thailand | HBA1:c.223G>C | 46 (12.4%) | 2.5±1.3 | 13.4±14.3 | 29.5±6.1 | 21 (45.7%) | 25 (54.3%) | 34 (73.9%) | 12 (26.1%) |
| Hb Q-India | HBA1:c.193G>C | 7 (1.9%) | 2.1±0.2 | 0 | 17.7±2.5 | 7 (100%) | 0 (0%) | 7 (100%) | 0 (0%) |
| Hb Hekinan | HBA1:c.84G>C | 6 (1.6%) | 2.0±0.6 | 0.3±0.2 | 36.6±3.3$^c$ | 6 (100%) | 0 (0%) | 5 (83.3%) | 1 (16.7%) |
| Hb Siam | HBA1:c.46G>C | 4 (1.1%) | 2.4±0.2 | 0 | 20.0±1.3 | 4 (100%) | 0 (0%) | 4 (100%) | 0 (0%) |
| Hb J-Wenchang-Wuming | HBA1:c.34A>C | 3 (0.8%) | 1.8±0.2 | 0 | 29.6±2.6 | 3 (100%) | 0 (0%) | 1 (33.3%) | 2 (66.7%) |
| Hb Thailand | HBA1:c.170A>C | 2 (0.5%) | 2.0, 2.1 | 0 | 21.2, 22.6 | 2 (100%) | 0 (0%) | 2 (100%) | 0 (0%) |
| Hb Grey Lynn | HBA1:c.274C>T | 1 (0.3%) | 2.4 | 0.5 | 15.4$^c$ | 1 (100%) | 0 (0%) | 1 (100%) | 0 (0%) |
| Hb J-Buda | HBA1:c.186G>T | 1 (0.3%) | 2.2 | 0 | 18.2$^c$ | 1 (100%) | 0 (0%) | 1 (100%) | 0 (0%) |
| **Total** | | **372 (100%)** | **3.1±1.1** | **6.5±12.8** | **33.9±11.9** | **211 (56.7%)** | **161 (43.3%)** | **257 (69.1%)** | **115 (30.9%)** |

a Hb F level and/or Hb variants co-migration with Hb F.

b Hb analysis using capillary electrophoresis.

c Hb analysis using HPLC.

performed step-by-step, including initial screening using a combined complete blood count and dichlorophenol indophenol precipitation (DCIP) test for Hb E (HBB:c.79G>A), followed by Hb analysis, and DNA analysis [3]. The molecular analysis of known mutations is typically performed using PCR-based methods, such as GAP-PCR, allele-specific PCR, reverse dot blot hybridization, and real-time PCR, often in the form of panels that include α⁰-thalassemia, α⁺-thalassemia, β-thalassemia, Hb variants, and high Hb F determinants [3,5,9]. While common α-thalassemia and β-thalassemia rarely pose diagnostic challenges, accurate diagnosis of Hb variants is sometimes difficult. Hb variants have a prevalence of around 2–3% in the Thai population and are less common than thalassemia [5–7]. In routine thalassemia diagnostics,

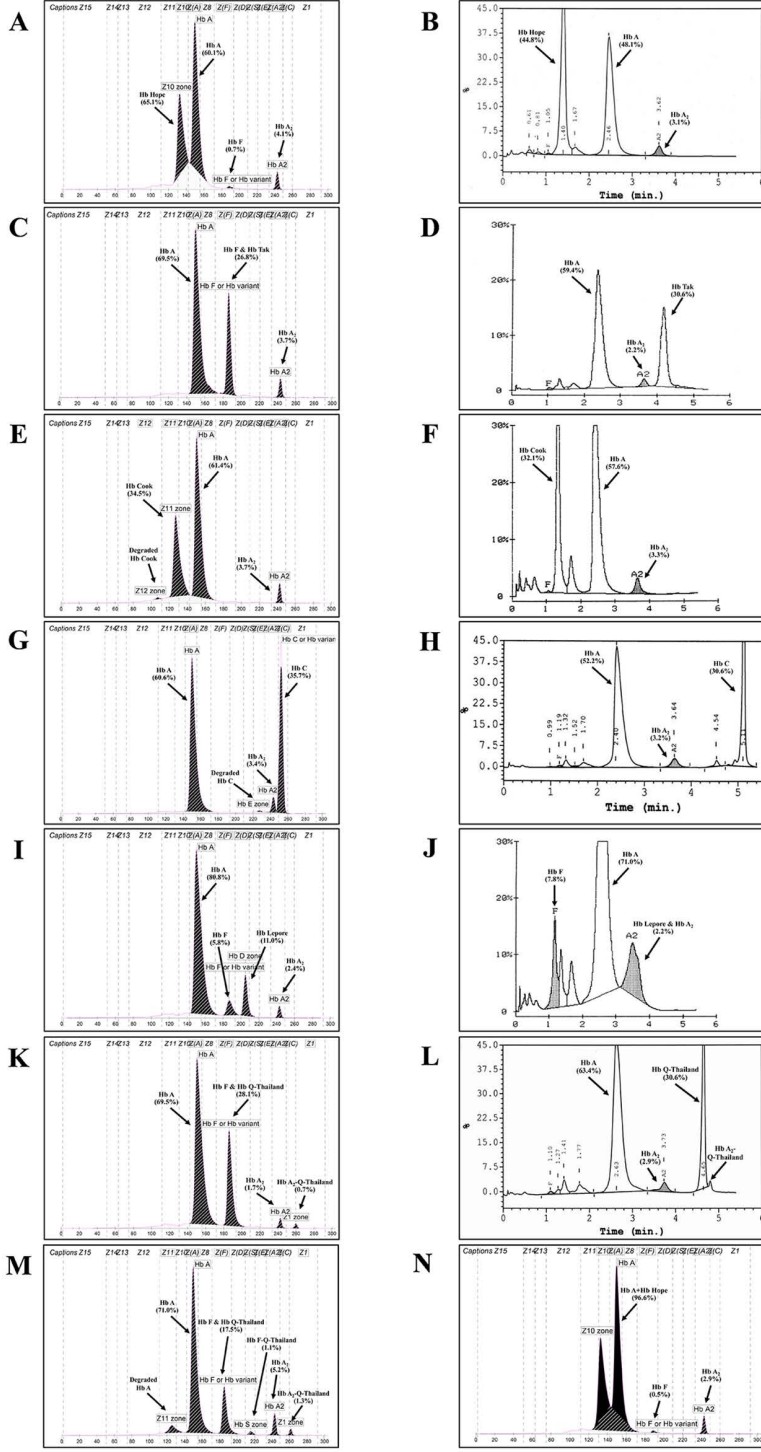

**Fig 2. Representative Hb analyses using capillary electrophoresis (A, C, E, G, I, K, M, and N) and HPLC (B, D, F, H, J, and L) in Hb variants with increased Hb A$_2$ levels and/or co-migration with Hb F.** These included heterozygote for Hb Hope **(A and B)**, the re-calculated Hb Hope **(N)**, Hb Tak **(C and D)**, Hb Cook **(E and F)**, Hb C **(G and H)**, Hb Lepore **(I and J)**, and Hb Q-Thailand **(K and L)**, and a double heterozygous Hb Q-Thailand and β-thalassemia **(M)**.

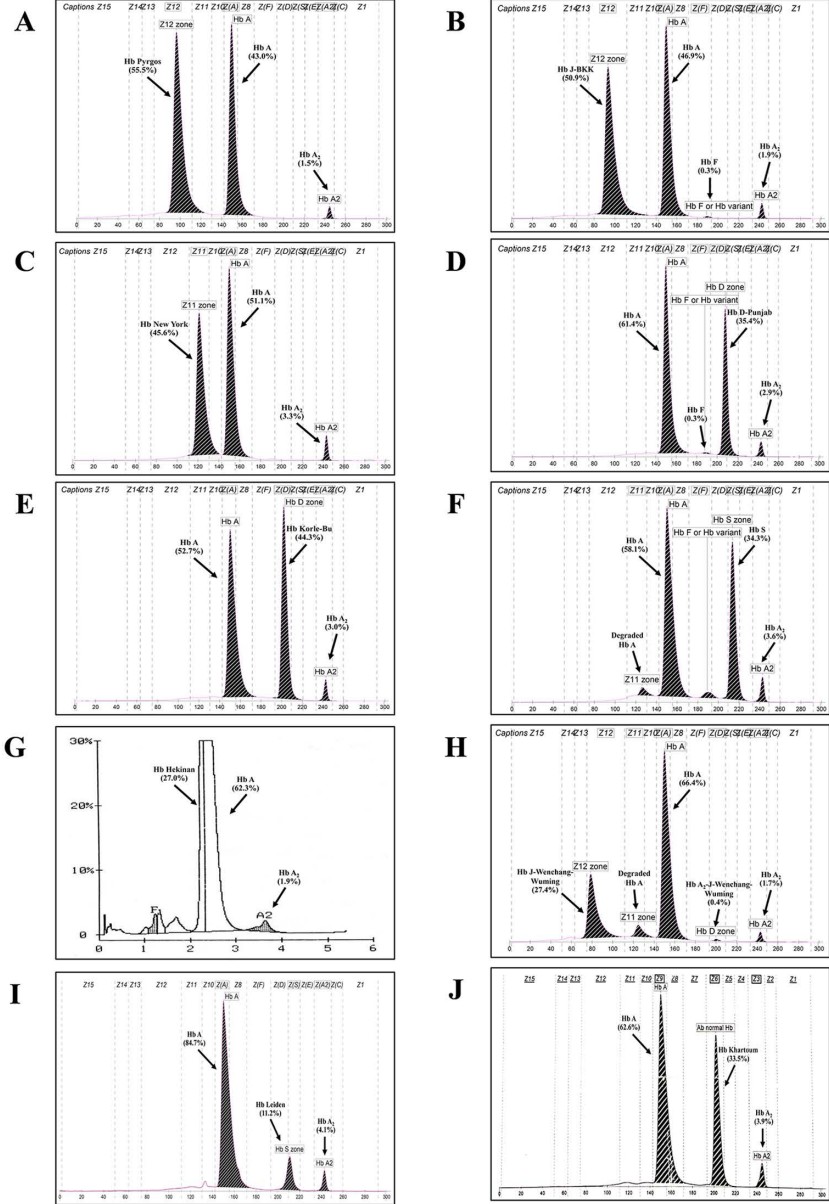

**Fig 3. Representative Hb analyses of Hb variants with low proportions of the request for β-thalassemia, including heterozygous Hb New York (A), Hb D-Punjab (B), Hb Korle-Bu (C), Hb S (D), Hb Hekinan (E), Hb J-Wenchang-Wuming (F), Hb Leiden (G), and Hb Khartoum (H).**

these Hb variants can lead to a misinterpretation of thalassemia and unnecessary prenatal diagnosis [8]. Here, we recorded the effects of Hb variants with increased Hb $A_2$ and/or co-migration with Hb F on the diagnosis of β-thalassemia in the Thai population.

The frequency of Hb variants found in this study during 2013–2025 (**Fig 1**) is quite similar to that of our previous study during 2002–2012 [5]. Hb Hope, Hb Tak, and Hb Q-Thailand, related to elevated Hb $A_2$ levels or co-migrated with Hb F levels, are the most common Hb variants. An elevated Hb $A_2$ level is an important marker for diagnosing β-thalassemia carriers [11–13]. Likewise, increased Hb F is associated with β-thalassemia disease and HPFH [1,2,9,14]. It

is conceivable, therefore, that an Hb variant with increased Hb $A_2$ and/or co-migrating with Hb F could be accompanied by an error of β-thalassemia or HPFH. Hb variants associated with elevated Hb $A_2$ have been occasionally described due to many factors, including unstable Hbs, Hb variant co-eluted or migrated with Hb $A_2$, and interference from analytical techniques, etc. [11–13]. However, information on the association of Hb variants with increased Hb $A_2$ and/or co-migrating with Hb F affecting the diagnosis of β-thalassemia is limited. Evidence is provided in this study, as shown in Table 1. It is noted that Hb variants with increased Hb $A_2$ levels and/or co-migrating with Hb F are not necessarily a marker of β-thalassemia trait, HFPH, or severe β-thalassemia diseases, especially the highest risk [71.9% (95% CI = 56.3–87.5) and odds ratio of 18.2 (7.1–48.9)] of the error in Hb variants with increased Hb $A_2$ and Hb F levels. Among these Hb variants, Hb Hope, Hb Tak, Hb Cook, Hb C, Hb Lepore, and Hb Q-Thailand are identified with increased Hb $A_2$ and/or Hb F levels, and a high proportion of the error. Hb Tak demonstrated the highest proportion of the error (61.2%) as it is associated with both elevated Hb $A_2$ levels and co-migration with Hb F in capillary electrophoresis. It is noteworthy, however, that although Hb Tak (Fig 2C) and Hb Q-Thailand (Fig 2K) co-migrate with Hb F on capillary electrophoresis, these two Hb variants could be separated from normal Hbs on Hb-HPLC analysis (Figs 2D,2L) [5]. Hb Tak, Hb Cook, Hb C, and Hb S are associated with borderline Hb $A_2$ levels (Table 2), unstable Hb, and hemolytic anemia [1,2,4,5,11,15–18]. In this study, other Hb variants with these phenotypes found for the first time in Thai subjects included Hb Leiden (HBB:c.22_24del-GAG) (Hb $A_2$ of 4.1%) (Fig 3I) and Hb Khartoum (HBB:c.374C > G) (Hb $A_2$ of 3.9%) (Fig 3J) [18,19]. It has been noted that Hb Hope is associated with falsely elevated Hb $A_2$ in capillary electrophoresis due to the presence of an uncalculated area between Hb Hope and Hb A (Fig 2A) [20]. The Hb $A_2$ levels of heterozygous Hb Hope in our series were falsely detected at 4.3 ± 0.5% (n = 35). However, after re-calculation incorporating the uncalculated area (Fig 2N), the corrected Hb $A_2$ levels were 2.8 ± 0.3%. Accordingly, this should be an important practice in routine thalassemia diagnostics to reduce the risk of misdiagnosis as β-thalassemia.

Next, Hb Lepore, a hybrid δβ-Hb variant, is co-eluted with Hb $A_2$ in HPLC and is associated with a slightly increased Hb F (Fig 2J). Misinterpretation of this Hb variant as a β-thalassemia on Hb-HPLC analysis is therefore common. However, this is not the case for capillary electrophoresis, where it is clearly separated from Hb $A_2$ [21]. Therefore, the error was observed in Hb variants with increased Hb $A_2$ levels and/or co-migration with Hb F, including Hb Hope, Hb Tak, Hb Cook, Hb C, Hb Lepore, and Hb Q-Thailand. Understanding the characteristics of these Hb variants, especially in prenatal thalassemia screening, would significantly reduce unnecessary prenatal diagnosis [8]. This diagnostic difficulty was not observed for Hb S, a well-known Hb variant.

Theoretically, the finding of Hb $A_2$A with Hb $A_2'$ and a Hb variant peak with the amount of around 25% should indicate an α-globin chain variant. In this case with Hb $A_2'$, it is recommended to report a total Hb $A_2$ level by combining Hb $A_2'$ with Hb $A_2$ in order not to misdiagnose a β-thalassemia carrier. This is exemplified by a subject with double heterozygosity for Hb Q-Thailand and β-thalassemia (Fig 2M). In contrast, Hb type of $A_2$A with a single Hb variant, with the amount of around 50% points to a β-globin chain variant. The compound heterozygote of this β-globin chain variant and β-thalassemia should reveal only Hb $A_2$ and Hb variant without Hb A. Thus, no β-thalassemia is expected in a subject with Hb $A_2$A with a single Hb variant, although accompanied by elevated Hb $A_2$. Hb E, the common β-Hb chain variant in Southeast Asia, is the best example. Hb analysis of heterozygous Hb E using capillary electrophoresis would reveal $A_2$EA with borderline Hb $A_2$ of 3.8 ± 0.3% without β-thalassemia [22].

Fig 4 demonstrated that, in many cases, an accurate diagnosis could be obtained using a simple Hb F cell staining test. Unlike Hb A, Hb F is not denatured by acid and can be stained by Amido Black B, which could be observable under a microscope, where red blood cells containing Hb A or other Hbs, which are denatured in an acid solution, would appear as ghost cells [23,24]. Hb Tak and Hb Q-Thailand are good examples. These two Hb variants co-migrate with Hb F on capillary electrophoresis and are easily misidentified as Hb F. Both are acid-sensitive and can be eluted from red blood cells. The Hb F cell staining test should detect no Hb F cells and reveal the ghost cells. Therefore, it is recommended to perform this simple Hb F cell staining to confirm the presence of the Hb F peak in Hb analysis before proceeding with specific DNA

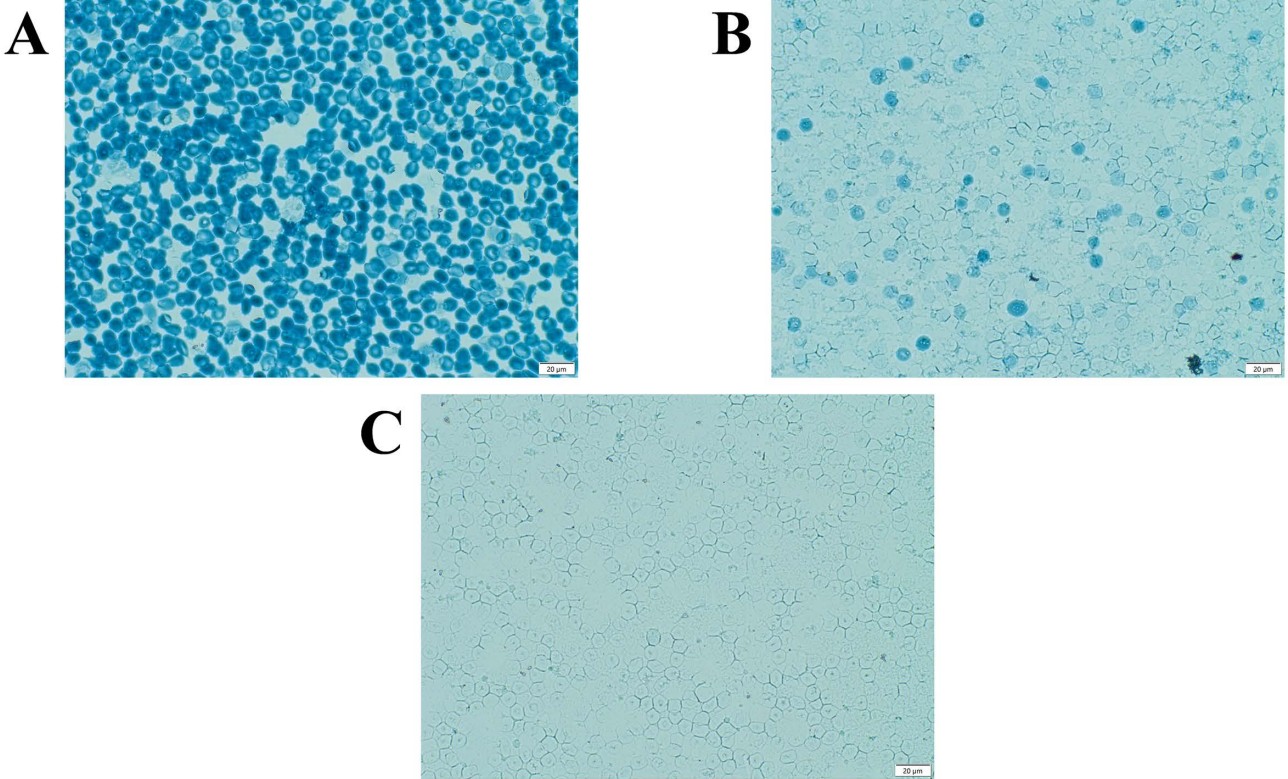

**Fig 4. Hb F cell staining by the simple acid elution test of the representative of a normal newborn with Hb F cells positive for 100% (A), an Hb E-β⁺-thalassemia with Hb F cells about 30% (B), and a heterozygous Hb Tak with Hb F cells less than 1% (C).**

analysis of the Hb variants. Alternatively, combined Hb analysis using two different assays, such as HPLC and capillary electrophoresis, should help confirm the presence of Hb F and Hb variants [5].

In conclusion, this study demonstrates that Hb variants with increased Hb $A_2$ levels and/or co-migration with Hb F, including Hb Hope, Hb Tak, Hb Cook, Hb C, Hb Lepore, and Hb Q-Thailand, are associated with a high proportion of unnecessary requests for β-thalassemia and HFPH. Interpretation using basic knowledge of Hb synthesis, combined with two different methods of Hb analysis and Hb F cell staining, should help resolve most diagnostic difficulties in a routine setting. This should prove useful in reducing errors and unnecessary prenatal thalassemia diagnoses, as well as reducing the workload of routine DNA analysis, thereby improving genetic counseling and management of a prevention and control program in the region.

## Supporting information

**S1 Data. Raw data used in Tables 1, 2 and Fig 1.**
(XLS)

## Author contributions

**Conceptualization:** Kritsada Singha, Goonnapa Fucharoen, Supan Fucharoen.

**Formal analysis:** Kritsada Singha, Hataichanok Srivorakun, Supawadee Yamsri, Attawut Chaibunruang, Anupong Pansuwan, Yossombat Changtrakul, Goonnapa Fucharoen, Supan Fucharoen.

**Funding acquisition:** Kritsada Singha, Supan Fucharoen.

**Investigation:** Kritsada Singha, Hataichanok Srivorakun, Supawadee Yamsri, Attawut Chaibunruang, Anupong Pansuwan, Yossombat Changtrakul.

**Methodology:** Kritsada Singha, Hataichanok Srivorakun, Supawadee Yamsri, Attawut Chaibunruang, Anupong Pansuwan, Yossombat Changtrakul.

**Project administration:** Kanokwan Sanchisuriya, Goonnapa Fucharoen, Supan Fucharoen.

**Resources:** Kanokwan Sanchisuriya, Goonnapa Fucharoen.

**Supervision:** Supan Fucharoen.

**Writing – original draft:** Kritsada Singha.

**Writing – review & editing:** Kritsada Singha, Hataichanok Srivorakun, Supawadee Yamsri, Attawut Chaibunruang, Anupong Pansuwan, Yossombat Changtrakul, Kanokwan Sanchisuriya, Goonnapa Fucharoen, Supan Fucharoen.

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
