## [Decision Letter · Decision Letter 0]

17 Feb 2026

Dear Dr. Fucharoen,

Thank you for submitting your manuscript to PLOS ONE. After careful consideration, we feel that it has merit but does not fully meet PLOS ONE’s publication criteria as it currently stands. Therefore, we invite you to submit a revised version of the manuscript that addresses the points raised during the review process.

We look forward to receiving your revised manuscript.

Kind regards,

J Francis Borgio, Ph.D.,

Academic Editor

PLOS One

Journal Requirements:

“This study was financially supported by Khon Kaen University, Thailand, to SF (Contract ID: RP68-Research Center KKU), and Genomics Thailand, the Health System Research Institute (Contract ID: HSRI 68-049) to KrS.”

3. We note that your Data Availability Statement is currently as follows: “All relevant data are within the manuscript files.”

Please confirm at this time whether or not your submission contains all raw data required to replicate the results of your study. Authors must share the “minimal data set” for their submission. PLOS defines the minimal data set to consist of the data required to replicate all study findings reported in the article, as well as related metadata and methods (https://journals.plos.org/plosone/s/data-availability#loc-minimal-data-set-definition).).).).

Reviewers' comments:

Reviewer's Responses to Questions

**Comments to the Author**

1. Is the manuscript technically sound, and do the data support the conclusions?

Reviewer #1: Partly

Reviewer #2: Yes

Reviewer #3: Yes

2. Has the statistical analysis been performed appropriately and rigorously?

Reviewer #1: No

Reviewer #2: N/A

Reviewer #3: Yes

3. Have the authors made all data underlying the findings in their manuscript fully available?

Reviewer #1: No

Reviewer #2: Yes

Reviewer #3: Yes

4. Is the manuscript presented in an intelligible fashion and written in standard English?

Reviewer #1: Yes

Reviewer #2: Yes

Reviewer #3: Yes

Reviewer #1: The work presented is technically sound. However, the phenomenon of Hemoglobin variants interfering with HbA2 quantification (leading to false-positive beta-thalassemia diagnosis) is well-documented in the literatures.

Major comments:

1. It is not quite clear how 372 were selected. Were these based on availability of the samples related to problematic reports of the hemoglobin variants? Or were these randomly selected from a larger pool of samples? Please clarify. The selection criteria could impact the generalizability of the findings.

2. The manuscript as currently presented does not provide sufficient novelty to warrant publication. The interference of hemoglobin variants on HbA2 quantification has been reported in multiple prior studies. The authors should better highlight what new information this study provides that is not already known in the field.

Minor comments:

1. The title which emphasize 12-year experience may not be necessary as the data chosen for analysis did not show how the trends evolved over the years. Consider revising the title to better reflect the content of the manuscript.

2. When presenting the prevalence of different hemoglobin variants, consider including the years the samples were collected to provide context on whether certain variants were more common in specific time periods. This should help strengthen the epidemiological aspect of the study.

3. The discussion section could benefit from a more in-depth comparison with prior studies on the same topic. Highlighting similarities and differences in findings would help situate this work within the broader literature.

4. The images included in the draft manuscript at present could be improved to have a better resolution when published, e.g. Fig 1 texts within the pictures are not legible.

5. The authors mentioned that the all data have been presented in the paper. However, as the aggregate data with mean, median, sd only. Individual levels data without PII could also be beneficials for future interests such as correlating specific type of mutation with the percentage of hemogobin electrophoresis results.

Reviewer #2: The authors emphasize that in their publications, the use of two different methods to measure Hb A2 and F values in the diagnosis of hemoglobinopathies, together with molecular analysis, highlights an important distinction in the interpretation of beta thalassemia. In this sense, they provide valuable data.

In some places in the text, the spelling of “Hb A2” varies. Consistency in spelling throughout the text may be preferable.

Reviewer #3: The manuscript addresses a well-chosen topic that offers a practical solution to a genuine laboratory challenge. However, I have a concern regarding the study population. In Thailand, Hb E is the most common hemoglobin variant and is known to be associated with elevated Hb A2 levels. Given the stated objectives of the study, it would be reasonable to include Hb E in the analysis. Nevertheless, this variant appears to have been excluded, and the authors did not provide a clear rationale for its omission. I would kindly request the authors to clarify why Hb E was not considered in the present study.

.

Reviewer #1: No

Reviewer #2: No

Reviewer #3: No

---

## [Author Response · Author response to Decision Letter 1]

3 Mar 2026

Responses to the academic editor

Journal Requirements:

Authors: We have prepared the revised manuscript to fit with the PLOS ONE style.

“This study was financially supported by Khon Kaen University, Thailand, to SF (Contract ID: RP68-Research Center KKU), and Genomics Thailand, the Health System Research Institute (Contract ID: HSRI 68-049) to KrS.”

Authors: We have stated the role of the funders in the revised manuscript under the section Research Funding.

3. We note that your Data Availability Statement is currently as follows: “All relevant data are within the manuscript files.”

Authors: Some raw data have been used in Tables 1 and 2 of the revised manuscript. The whole raw data file is submitted as a supplementary file at this revision.

Authors: There is no recommendation from the reviewers to cite specific previously published works.

Response to reviewers comments

Reviewer #1: The work presented is technically sound. However, the phenomenon of Hemoglobin variants interfering with HbA2 quantification (leading to false-positive beta-thalassemia diagnosis) is well-documented in the literatures.

Major comments:

1. It is not quite clear how 372 were selected. Were these based on availability of the samples related to problematic reports of the hemoglobin variants? Or were these randomly selected from a larger pool of samples? Please clarify. The selection criteria could impact the generalizability of the findings.

Authors: As mentioned in the Materials and Methods section on page 4, lines 78-83 of the revised manuscript, we looked retrospectively on a total of 43,414 subjects referred to our center for investigation of thalassemia and hemoglobinopathies from 01/01/2013 to 31/07/2025. A total of 372 subjects with Hb variants in heterozygotic form were selectively recruited. All subjects with Hb variants but Hb E were included. This is because it has been known that Hb E, the most common Hb variant in Southeast Asia, is associated with elevated Hb A2 and can be easily diagnosed on Hb analysis [2,3].

2. The manuscript as currently presented does not provide sufficient novelty to warrant publication. The interference of hemoglobin variants on HbA2 quantification has been reported in multiple prior studies. The authors should better highlight what new information this study provides that is not already known in the field.

Authors: In the revised version, Discussion related to this has been provided additionally on page 11, lines 223-228, highlighting related new information, as recommended.

Minor comments:

1. The title which emphasize 12-year experience may not be necessary as the data chosen for analysis did not show how the trends evolved over the years. Consider revising the title to better reflect the content of the manuscript.

Authors: In the revised version, the title has been changed to … “Common hemoglobin variants affecting the diagnosis of β-thalassemia: a large cohort data at a single center” …, as recommended.

2. When presenting the prevalence of different hemoglobin variants, consider including the years the samples were collected to provide context on whether certain variants were more common in specific time periods. This should help strengthen the epidemiological aspect of the study.

Authors: In the revision, we have created a new Figure 1 to present the number of different Hb variants encountered each year of specimen collection which was mentioned in the Results section (page 5, lines 108-109) of the revised manuscript, as recommended. Figure numbers are changed accordingly.

3. The discussion section could benefit from a more in-depth comparison with prior studies on the same topic. Highlighting similarities and differences in findings would help situate this work within the broader literature.

Authors: In the revised version, additional discussion comparing with prior studies was provided on page 11, lines 218-221, and 224-228, as recommended.

4. The images included in the draft manuscript at present could be improved to have a better resolution when published, e.g. Fig 1 texts within the pictures are not legible.

Authors: At the revision, all images have been improved to have higher resolution as recommended.

5. The authors mentioned that the all data have been presented in the paper. However, as the aggregate data with mean, median, sd only. Individual levels data without PII could also be beneficials for future interests such as correlating specific type of mutation with the percentage of hemoglobin electrophoresis results.

Authors: In the revised version, we have provided percentage of Hb variants in Table 2. The whole raw data is also submitted as a supplementary file.

Reviewer #2: The authors emphasize that in their publications, the use of two different methods to measure Hb A2 and F values in the diagnosis of hemoglobinopathies, together with molecular analysis, highlights an important distinction in the interpretation of beta thalassemia. In this sense, they provide valuable data. In some places in the text, the spelling of “Hb A2” varies. Consistency in spelling throughout the text may be preferable.

Authors: We thank the reviewer to bring this typing error to our attention. The revised manuscript has been checked thoroughly for the consistency in spelling before submission.

Reviewer #3: The manuscript addresses a well-chosen topic that offers a practical solution to a genuine laboratory challenge. However, I have a concern regarding the study population. In Thailand, Hb E is the most common hemoglobin variant and is known to be associated with elevated Hb A2 levels. Given the stated objectives of the study, it would be reasonable to include Hb E in the analysis. Nevertheless, this variant appears to have been excluded, and the authors did not provide a clear rationale for its omission. I would kindly request the authors to clarify why Hb E was not considered in the present study.

Authors: Hb E is the most common Hb variant in Southeast Asia. In Thailand, the prevalence of Hb E ranges from 10% to 53% across the regions. Because of its high prevalence and well-known characteristic of elevated Hb A2, Hb E can be diagnosed easily on Hb analysis without affecting the diagnosis of β-thalassemia in this country. We therefore excluded Hb E in this study. This has been mentioned on page 4, lines 81-83 of the revised manuscript.

---

## [Decision Letter · Decision Letter 1]

24 Mar 2026

Common hemoglobin variants affecting the diagnosis of β-thalassemia: a large cohort data at a single center

PONE-D-25-66507R1

Dear Dr. Fucharoen,

We’re pleased to inform you that your manuscript has been judged scientifically suitable for publication and will be formally accepted for publication once it meets all outstanding technical requirements.

Kind regards,

J Francis Borgio, Ph.D.,

Academic Editor

PLOS One

Reviewers' comments:

Reviewer's Responses to Questions

**Comments to the Author**

Reviewer #1: All comments have been addressed

Reviewer #2: All comments have been addressed

2. Is the manuscript technically sound, and do the data support the conclusions?

Reviewer #1: Yes

Reviewer #2: Yes

3. Has the statistical analysis been performed appropriately and rigorously?

Reviewer #1: Yes

Reviewer #2: I Don't Know

4. Have the authors made all data underlying the findings in their manuscript fully available?

Reviewer #1: Yes

Reviewer #2: Yes

5. Is the manuscript presented in an intelligible fashion and written in standard English?

Reviewer #1: Yes

Reviewer #2: Yes

Reviewer #1: I have reviewed the revised manuscript (PONE-D-25-66507R1) and the authors' corresponding rebuttal letter. The authors have made sufficient efforts to address the primary concerns raised during the initial review, particularly regarding the clarification of their inclusion criteria and the commendable provision of the underlying dataset. While the manuscript remains predominantly descriptive and misses an opportunity to employ more advanced statistical modeling—which could have substantially elevated its diagnostic impact—the data presented are technically sound and offer practical value to the field of hematological screening. The current revisions meet the threshold for publication, and I therefore recommend that the manuscript be accepted in its current form.

Reviewer #2: The authors have been done all the comments from the reviewers. I think this is sufficient for publish.

.

Reviewer #1: No

Reviewer #2: No

---

## [Editor Report · Acceptance letter]

PONE-D-25-66507R1

PLOS One

Dear Dr. Fucharoen,

I'm pleased to inform you that your manuscript has been deemed suitable for publication in PLOS One. Congratulations! Your manuscript is now being handed over to our production team.

Kind regards,

on behalf of

Dr. J Francis Borgio

Academic Editor

PLOS One